# CARLANE: A Lane Detection Benchmark for Unsupervised Domain Adaptation from Simulation to multiple Real-World Domains

**Julian Gebele**[*,1]  **Bonifaz Stuhr**[*,1,2]  **Johann Haselberger**[*,1,3]
[1]University of Applied Science Kempten
[2]Autonomous University of Barcelona
[3]Technische Universität Berlin
carlane.benchmark@gmail.com

## Abstract

Unsupervised Domain Adaptation demonstrates great potential to mitigate domain shifts by transferring models from labeled source domains to unlabeled target domains. While Unsupervised Domain Adaptation has been applied to a wide variety of complex vision tasks, only few works focus on lane detection for autonomous driving. This can be attributed to the lack of publicly available datasets. To facilitate research in these directions, we propose CARLANE, a 3-way sim-to-real domain adaptation benchmark for 2D lane detection. CARLANE encompasses the single-target datasets MoLane and TuLane and the multi-target dataset MuLane. These datasets are built from three different domains, which cover diverse scenes and contain a total of 163K unique images, 118K of which are annotated. In addition we evaluate and report systematic baselines, including our own method, which builds upon Prototypical Cross-domain Self-supervised Learning. We find that false positive and false negative rates of the evaluated domain adaptation methods are high compared to those of fully supervised baselines. This affirms the need for benchmarks such as CARLANE to further strengthen research in Unsupervised Domain Adaptation for lane detection. CARLANE, all evaluated models and the corresponding implementations are publicly available at **https://carlanebenchmark.github.io**.

## 1 Introduction

Vision-based deep learning systems for autonomous driving have made significant progress in the past years [1–5]. Recent state-of-the-art methods achieve remarkable results on public, real-world benchmarks but require labeled, large-scale datasets. Annotations for these datasets are often hard to acquire, mainly due to the high expenses of labeling in terms of cost, time, and difficulty. Instead, simulation environments for autonomous driving, such as CARLA [6], can be utilized to generate abundant labeled images automatically. However, models trained on data from simulation often experience a significant performance drop in a different domain, i.e., the real world, mainly due to the domain shift [7]. Unsupervised Domain Adaptation (UDA) methods [8–15] try to mitigate the domain shift by transferring models from a fully-labeled source domain to an unlabeled target domain. This eliminates the need for annotating images but assumes that the target domain is accessible at training time. While UDA has been applied to complex tasks for autonomous driving such as object detection [1, 16] and semantic segmentation [17, 18], only few works focus on lane detection [19, 5]. This can be attributed to the lack of public UDA datasets for lane detection.

---

[*]Equal contribution

36th Conference on Neural Information Processing Systems (NeurIPS 2022) Track on Datasets and Benchmarks.

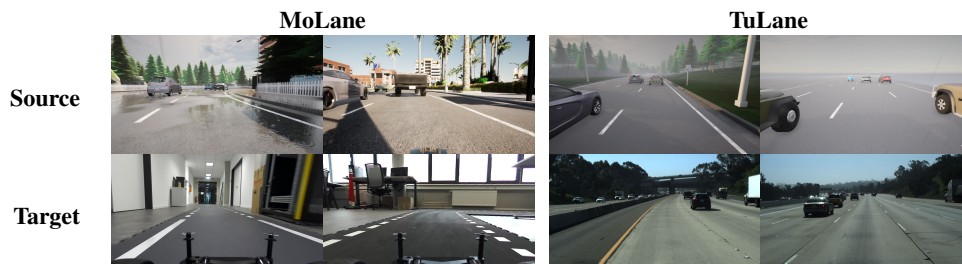

Figure 1: Images sampled from our CARLANE Benchmark.

To compensate for this data scarcity and encourage future research, we introduce CARLANE, a sim-to-real domain adaptation benchmark for lane detection. We use the CARLA simulator for data collection in the source domain with a free-roaming waypoint-based agent and data from two distinct real-world domains as target domains. This enables us to construct a benchmark that consists of three datasets:

*(1) MoLane* focuses on abstract lane markings in the domain of a 1/8th *Mo*del vehicle. We collect 80K labeled images from simulation as the source domain and 44K unlabeled real-world images from several tracks with two lane markings as the target domain. Further, we apply domain randomization as well as data balancing. For evaluation, we annotate 2,000 validation and 1,000 test images with our labeling tool.

*(2) TuLane* incorporates 24K balanced and domain-randomized images from simulation as the source domain and the well-known *Tu*Simple [20] dataset with 3,268 real-world images from U.S. highways with up to four labeled lanes as the target domain. The target domain of MoLane is a real-world abstraction from the target domain of TuLane, which may result in interesting insights about UDA.

*(3) MuLane* is a balanced combination of *Mo*Lane and T*u*Lane with two target domains. For the source domain, we randomly sample 24K images from MoLane and combine them with TuLane's synthetic images. For the target domains, we randomly sample 3,268 images from MoLane and combine them with TuSimple. This allows us to investigate multi-target UDA for lane detection.

To establish baselines and investigate UDA on our benchmark, we evaluate several adversarial discriminative methods, such as DANN [12], ADDA [13] and SGADA [21]. Additionally, we propose SGPCS, which builds upon PCS [22] with a pseudo labeling approach to achieve state-of-the-art performance.

Our contributions are three-fold: (1) We introduce CARLANE, a 3-way sim-to-real benchmark, allowing single- and multi-target UDA. (2) We provide several dataset tools, i.e., an agent to collect images with lane annotations in CARLA and a labeling tool to annotate the real-world images manually. (3) We evaluate several well-known UDA methods to establish baselines and discuss results on both single- and multi-target UDA. To the best of our knowledge, we are the first to adapt a lane detection model from simulation to multiple real-world domains.

## 2 Related Work

### 2.1 Data Generation for Sim-to-Real Lane Detection

In recent years, much attention has been paid to lane detection benchmarks in the real world, such as CULane [3], TuSimple [20], LLAMAS [23], and BDD100K [24]. Despite the popularity of these benchmarks, there is few research that focuses on sim-to-real lane detection datasets. Garnett et al. [4] propose a method for generating synthetic images with 3D lane annotations in the open-source engine blender. Their *synthetic-3D-lanes* dataset contains 300K train, 1,000 validation and 5,000 test images, while their real-world *3D-lanes* dataset consists of 85K images, which are annotated in a semi-manual manner. Utilizing the data generation method from [4], Garnett et al. [19] collect 50K labeled synthetic images to perform sim-to-real domain adaptation for 3D lane detection. At this point, the source domain of the dataset is not publicly available.

Recently, Hu et al. [5] investigated UDA techniques for 2D lane detection. Their proposed data generation method relies on CARLA's built-in agent to automatically collect 16K synthetic images.

However, the dataset is not publicly available at this point. In comparison, our method leverages an efficient and configurable waypoint-based agent. Furthermore, in contrast to the aforementioned works, considering only single-source single-target UDA, we additionally focus on multi-target UDA.

## 2.2 Unsupervised Domain Adaptation

Unsupervised Domain Adaptation has been extensively studied in recent years [9]. In an early work, Ganin et al. [8] propose a gradient reversal layer between the features extractor and a domain classifier to learn similar feature distributions for distinct domains. Early discrepancy-based methods employ a distance metric to measure the discrepancy of the source and target domain [10, 25]. A prominent example is DAN [10] which uses maximum mean discrepancies (MMD) [26, 27] to match embeddings of different domain distributions. Recently, DSAN [11] builds upon DAN with local MMD and exploits fine-grained features to align subdomains accurately.

Domain alignment can also be achieved through adversarial learning [28]. Adversarial discriminative methods such as DANN [12] or ADDA [13] employ a domain classifier or discriminator, encouraging the feature extractor to produce domain-invariant representations. While these methods mainly rely on feature-level alignment, adversarial generative methods [29, 30] operate on pixel-level.

In a recent trend, self-supervised learning methods are leveraged as auxiliary tasks to improve domain adaptation effectiveness and to capture in-domain semantic structures [14, 15, 31, 22, 32]. Furthermore, self-supervised learning is utilized for cross-domain alignment as well, by matching class-discriminative features [31, 33], task-discriminative features [34], class prototypes [35, 22] or equivalent samples in the domains [36].

Furthermore, other recent works mitigate optimization inconsistencies by minimizing the gradients discrepancy of the source samples and target samples [37] or by applying a meta-learning scheme between the domain alignment and the targeted classification task [38].

# 3 Data Generation

To construct our benchmark, we gather image data from a real 1/8th model vehicle, and the CARLA simulator [6]. Ensuring the verification of results and transferability to real driving scenarios, we extend our benchmark with the TuSimple dataset [20]. This enables gradual testing, starting from simulation, followed by model cars, and ending with full-scale real word experiments. Data variety is achieved through domain randomization in all domains. However, naively performing domain randomization might lead to an imbalanced dataset. Therefore, similar driving scenarios are sampled across all domains, and a bagging approach is utilized to uniformly collect lanes by their curvature with respect to the camera position. We strictly follow TuSimple's data format [20] to maintain consistency across all our datasets.

## 3.1 Real-World Environment

As shown in Figure 2, we build six different 1/8th race tracks, where each track is available in two different surface materials (dark and light gray). We vary between dotted and solid lane markings, which are pure white and $50\,\text{mm}$ thick. The lanes are constantly $750\,\text{mm}$ wide, and the smallest inner radius is $250\,\text{mm}$. The track layouts are designed to roughly contain the same proportion of straight and curved segments to obtain a balanced label distribution. We construct these tracks in four locations with alternating backgrounds and lighting conditions.

## 3.2 Real-World Data Collection

Raw image data is recorded from a front-facing Stereolabs ZEDM camera with 30 FPS and a resolution of $1280 \times 720$ pixels. A detailed description of the 1/8th car can be found in the Appendix. The vehicle is moved with a quasi-constant velocity clockwise and counter-clockwise to cover both directions of each track. All collected images from tracks (e) and (f) are used for the test subset. In addition, we annotate lane markings with our labeling tool for validation and testing, which is made publicly available.

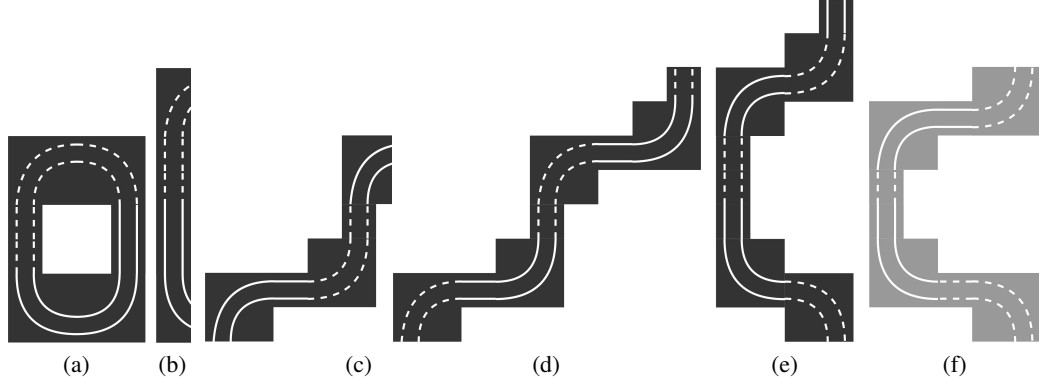

|     |     |     |     |     |     |
| :-: | :-: | :-: | :-: | :-: | :-: |
| (a) | (b) | (c) | (d) | (e) | (f) |

Figure 2: Overview of our track types for MoLane. (a) - (d) show the black version of the training and validation tracks. These tracks are also constructed using a light gray surface material. (e) and (f) depict our test tracks.

### 3.3 Simulation Environment

We utilize the built-in APIs from CARLA to randomize multiple aspects of the agent and environment, such as weather, daytime, ego vehicle position, camera position, distractor vehicles, and world objects (i.a., walls, buildings, and plants). Weather and daytime are varied systematically by adapting parameters for cloud density, rain intensity, puddles, wetness, wind strength, fog density, sun azimuth, and sun altitude. For further details, we refer to our implementation. To occlude the lanes similar to real-world scenarios, up to five neighbor vehicles are spawned randomly in the vicinity of the agent. We consider five different CARLA maps in urban and highway environments (Town03, Town04, Town05, Town06, and Town10) to collect our dataset, as the other towns' characteristics are not suitable for our task (i.a., mostly straight lanes). In addition, we collect data from the same towns without world objects to strengthen the focus on lane detection, similar to our model vehicle target domain.

### 3.4 Simulation Data Agent

We implement an efficient agent based on waypoint navigation, which roams randomly and reliably in the aforementioned map environments and collects $1280 \times 720$ images. In each step, the waypoint navigation stochastically traverses the CARLA road map with a fixed lookahead distance of one meter. In addition, we sample offset values $\Delta y_k$ from the center lane within the range $\pm 1.20\,\mathrm{m}$.

To avoid saturation at the lane borders, which would occur with a sinusoidal function, we use the triangle wave function:

$$\Delta y_k = \frac{2m}{\pi} \arcsin(\sin(i_k)) \tag{1}$$

where $m$ is the maximal offset and $i_k$ is incremented by $0.08$ for each simulation step $k$. Per frame, our agent moves to the next waypoint with an increment of one meter, enabling the collection of highly diverse data in a fast manner. We use a bagging approach for balancing, which allows us to define lane classes based on their curvature.

## 4 The CARLANE Benchmark

The CARLANE Benchmark consists of three distinct sim-to-real datasets, which we build from our three different domains. The details of the individual subsets can be found in Table 1.

*MoLane* consists of images from CARLA and the real 1/8th model vehicle. For the abstract real-world domain, we collect 46,843 images with our model vehicle, of which 2,000 validation and 1,000 test images are labeled. For the source domain, we use our simulation agent to gather 84,000 labeled images. To match the label distributions between both domains, we define five lane classes based on the relative angle $\beta$ of the agent to the center lane for our bagging approach: strong left curve

Table 1: Dataset overview. Unlabeled images denoted by *, partially labeled images denoted by **

| Dataset | domain | total images | train | validation | test | lanes |
|---|---|---|---|---|---|---|
| MoLane | CARLA simulation | 84,000 | 80,000 | 4,000 | - | $\leq 2$ |
|  | model vehicle | 46,843 | 43,843* | 2,000 | 1,000 | $\leq 2$ |
| TuLane | CARLA simulation | 26,400 | 24,000 | 2,400 | - | $\leq 4$ |
|  | TuSimple [20] | 6,408 | 3,268 | 358 | 2,782 | $\leq 4$ |
| MuLane | CARLA simulation | 52,800 | 48,000 | 4,800 | - | $\leq 4$ |
|  | model vehicle + TuSimple [20] | 12,536 | 6,536** | 4,000 | 2,000 | $\leq 4$ |

($\beta \leq -45°$), soft left curve ($-45° < \beta \leq -15°$), straight ($-15° < \beta < 15°$), soft right curve ($15° \leq \beta < 45°$) and strong right curve ($45° \leq \beta$). In total, MoLane encompasses 130,843 images.

*TuLane* consists of images from CARLA, and a cleaned version of the TuSimple dataset [20], which is licensed under the Apache License, Version 2.0. To clean test set annotations, we utilize our labeling tool to ensure that the up to four lanes closest to the car are correctly labeled. We adapt the bagging classes to align the source dataset with TuSimple's lane distribution: left curve ($-12°$ $< \beta \leq 5°$), straight ($-5° < \beta < 5°$) and right curve ($5° \leq \beta < 12°$).

*MuLane* is a multi-target UDA dataset and is a balanced mixture of images from MoLane and TuLane. For MuLane's entire training set and its source domain validation and test set, we use all available images from TuLane and sample the same amount of images from MoLane. We adopt the 1,000 test images from MoLane's target domain and sample 1,000 test images from TuSimple to form MuLane's test set. For the validation set, we use the 2,000 validation images from MoLane and 2,000 of the remaining validation and test images of TuLane's target domain. In total, MuLane consists of 65,336 images.

To further analyze CARLANE, we visualize the ground truth lane distributions in Figure 3. We observe that the lane distributions of source and target data from our datasets are well aligned.

MoLane, TuLane, and MuLane are publicly available at https://carlanebenchmark.github.io and licensed under the Apache License, Version 2.0.

### 4.1 Dataset Format

For each dataset, we split training, validation, and test samples into source and target subsets. Lane annotations are stored within a *.json* file containing the lanes' y-values discretized by raw anchors, the lanes' x-values, and the image file path following the data format of TuSimple[20]. Additionally, we adopt the method from [2] to generate *.png* lane segmentations and a *.txt* file containing the linkage between the raw images and their segmentation as well as the presence and absence of a lane.

### 4.2 Dataset Tasks

The main task of our datasets is UDA for lane detection, where the goal is to predict lane annotations $Y_t \in \mathbb{R}^{R \times G \times N}$ given the input image $X_t \in \mathbb{R}^{H \times W \times 3}$ from the unlabeled target domain $\mathcal{D}_\mathcal{T} = \{(X_t)\}_{t \in \mathcal{T}}$. $R$ defines the number of row anchors, $G$ the number of griding cells, and $N$ the number of lane annotations available in the dataset, where the definition of $Y_t$ follows [20]. During training time, the images $X_s \in \mathbb{R}^{H \times W \times 3}$, corresponding labels $Y_s \in \mathbb{R}^{H \times W \times C}$ from the source domain $\mathcal{D}_\mathcal{S} = \{(X_s, Y_s)\}_{s \in \mathcal{S}}$, and the unlabeled target images $X_t$ are available. Additionally, MuLane focuses on multi-target UDA, where $\mathcal{D}_\mathcal{T} = \{(X_{t_1}) \cup (X_{t_2})\}_{t_1 \in \mathcal{T}_1, t_2 \in \mathcal{T}_2}$.

Although we focus on sim-to-real UDA, our datasets can be used for unsupervised and semi-supervised tasks and partially for supervised learning tasks. Furthermore, a real-to-real transfer can be performed between the target domains of our datasets.

## 5 Benchmark Experiments

We conduct experiments on our CARLANE Benchmark for several UDA methods from the literature and our proposed method. Additionally, we train fully supervised baselines on all domains.

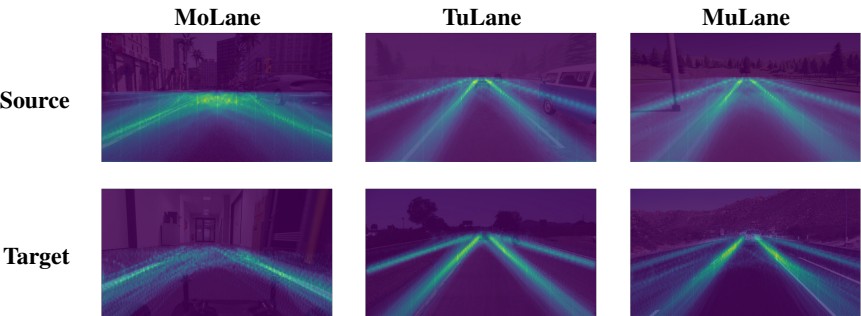

| MoLane | TuLane | MuLane |

Source

Target

Figure 3: Lane annotation distributions of the three subsets of CARLANE. Since the real-world training data of MoLane and MuLane is unlabeled, we utilize their validation data for visualization.

## 5.1 Metrics

For evaluation, we use the following metrics:

*(1) Lane Accuracy* (LA) [2] is defined by LA $= \frac{p_c}{p_y}$, where $p_c$ is the number of correctly predicted lane points and $p_y$ is the number of ground truth lane points. Lane points are considered as correct if their $L_1$ distance is smaller than the given threshold $t_{pc} = \frac{20}{\cos(a_{yl})}$, where $a_{yl}$ is the angle of the corresponding ground truth lane.

*(2) False Positives* (FP) and *False Negatives* (FN) [2]: To further determine the error rate and to draw more emphasis on mispredicted or missing lanes, we measure false positives with FP $= \frac{l_f}{l_p}$ and false negatives with FN $= \frac{l_m}{l_y}$, where $l_f$ is the number of mispredicted lanes, $l_p$ is the number of predicted lanes, $l_m$ is the number of missing lanes and $l_y$ is the number of ground truth lanes. Following [2], we classify lanes as mispredicted, if the $LA < 85\%$.

## 5.2 Baselines

We use Ultra Fast Structure-aware Deep Lane Detection (UFLD) [2] as baseline and strictly adopt its training scheme and hyperparameters. UFLD treats lane detection as a row-based classification problem and utilizes the row anchors defined by TuSimple [20]. To achieve a lower bound for the evaluated UDA methods, we train UFLD as a supervised baseline on the source simulation data (UFLD-SO). Furthermore, we train our baseline on the labeled real-world training data for a surpassable fully-supervised performance in the target domain (UFLD-TO). Since the training images from MoLane and MuLane have no annotations, we train UFLD-TO in these cases on the labeled validation images and validate our model on the entire test set.

## 5.3 Compared UDA Methods

We evaluate the following feature-level UDA methods on the CARLANE Benchmark by adopting their default hyperparameters and tuning them accordingly. Each model is initialized with the pre-trained feature encoder of our baseline model (UFLD-SO). The optimized hyperparameters can be found in Table 2.

*(1) DANN* [12] is an adversarial discriminative method that utilizes a shared feature encoder and a dense domain classifier connected via a gradient reversal layer.

*(2) ADDA* [13] employs a feature encoder for each domain and a dense domain discriminator. Following ADDA, we freeze the weights of the pre-trained classifier of UFLD-SO to obtain final predictions.

*(3) SGADA* [21] builds upon ADDA and utilizes its predictions as pseudo labels for the target training images. Since UFLD treats lane detection as a row-based classification problem, we reformulate the pseudo label selection mechanism. For each lane, we select the highest confidence value from the griding cells of each row anchor. Based on their griding cell position, the confidence values are divided into two cases: absent lane points and present lane points. Thereby, the last griding cell

Table 2: Optimized hyperparameters to achieve the reported results. $C$ denotes domain classifier parameters, $D$ denotes domain discriminator parameters, adv the adversarial loss from [13] and cls the classifier loss, sim the similarity loss and aux the auxiliary loss from [2]. Loss weights are set to 1.0 unless stated otherwise.

| Method | Initial Learning Rate | Scheduler | Batch Size | Epochs | Losses | Other Changes |
|---|---|---|---|---|---|---|
| UFLD-SO | $4e^{-4}$ | Cosine Annealing | 4 | 150 | cls, sim, aux | - |
| DANN | $1e^{-5}, C: 1e^{-3}$ | $\frac{1e^{-5}}{(1+10p)^{0.75}}$ | 4 | 30 | cls, sim, aux, adv [12] | $C$: 3 fc layers (1024-1024-2) |
| ADDA | $1e^{-6}, D: 1e^{-3}$ | Constant | 16 | 30 | map [13], adv [13] | $D$: 3 fc layers (500-500-2) |
| SGADA | $1e^{-6}, D: 1e^{-3}$ | Constant | 16 | 15 | map [13], adv [13], pseudo: 0.25 | Pseudo label selection |
| SGPCS | $4e^{-4}$ | Cosine Annealing | 16 | 10 | in-domain [22], cross-domain [22] cls, sim, aux, pseudo: 0.25 | - |
| UFLD-TO | $4e^{-4}$ | Cosine Annealing | 4 | 300 | cls, sim, aux | - |

represents absent lane points as in [2]. For each case, we calculate the mean confidence over the corresponding lanes. We then use the thresholds defined by SGADA to decide whether the prediction is treated as a pseudo label.

*(4) SGPCS* (ours) builds upon PCS [22] and performs in-domain contrastive learning and cross-domain self-supervised learning via cluster prototypes. Our overall objective function comprises the in-domain and cross-domain loss from PCS, the losses defined by UFLD, and our adopted pseudo loss from SGADA. We adjust the momentum for memory bank feature updates to $0.5$ and use spherical K-means [39] with $K = 2,500$ to cluster them into prototypes.

## 5.4 Implementation Details

We implement all methods in PyTorch 1.8.1 and train them on a single machine with four RTX 2080 Ti GPUs. Tuning all methods took a total amount of compute of approximately 3.5 petaflop/s-days. The training times for each model range from 4-13 days for UFLD baselines and 6-44 hours for domain adaption methods. In addition, we found that applying output scaling on the last linear layer of the model yields slightly better results. Therefore, we divide the models' output by 0.5. Our implementation is publicly available at https://carlanebenchmark.github.io.

## 5.5 Evaluation

**Quantitative Evaluation.** In Table 3 we report the results on MoLane, TuLane, and MuLane across five different runs. We observe that UFLD-SO is able to generalize to a certain extent to the target domain. This is mainly due to the alignment of semantic structure from both domains. ADDA, SGADA, and our proposed SGPCS manage to adapt the model to the target domain slightly and consistently. However, DANN suffers from negative transfer [40] when trained on MoLane and MuLane. The negative transfer of DANN for complex domain adaptation tasks is also observed in other works [35, 41, 40, 42] and can be explained by the source domain's data distribution and the model complexity [40]. In our case, the source domain contains labels not present in the target domain, as shown in Figure 3, which is more pronounced in MoLane and MuLane.

We want to emphasize that with an accuracy gain of a maximum of 4.55% (SGPCS) and high false positive and false negative rates, the domain adaptation methods are not able to achieve comparable results to the supervised baselines (UFLD-TO). Furthermore, we observe that false positive and false negative rates increase significantly on MuLane, indicating that the multi-target dataset forms the most challenging task. False positives and false negatives represent wrongly detected and missing lanes which can lead to crucial impacts on autonomous driving functions. These results affirm the need for the proposed CARLANE Benchmark to further strengthen the research in UDA for lane detection.

**Qualitative Evaluation.** We use t-SNE [43] to visualize the features of the features encoders for the source and target domains of MuLane in Figure 4. t-SNE visualizations of MoLane and TuLane can be found in the Appendix. In accordance with the quantitative results, we observe only a slight adaptation of the source and target domains features for ADDA, SGADA, and SGPCS compared to the supervised baseline UFLD-SO. Consequently, the examined well-known domain adaptation methods have no significant effect on feature alignment. In addition, we show results from the evaluated methods in Figure 5 and observe that the models are able to predict target domain lane

Table 3: Performance on the test set. Lane accuracy (LA), false positives (FP), and false negatives (FN) are reported in %.

| ResNet-18 | MoLane | | TuLane | | | MuLane | | |
|---|---|---|---|---|---|---|---|---|
| | LA | FP & FN | LA | FP | FN | LA | FP | FN |
| UFLD-SO | 89.39 | 25.25 | 87.43 | 34.21 | 23.48 | 88.02 | 50.24 | 26.08 |
| DANN [12] | 87.65±0.48 | 29.97±1.21 | 88.74±0.32 | 32.71±0.52 | 21.64±0.65 | 86.01±0.67 | 55.33±1.22 | 36.30±1.90 |
| ADDA [13] | 92.85±0.17 | 10.61±0.77 | 90.72±0.15 | 29.73±0.36 | 17.67±0.42 | 89.83±0.33 | 46.79±0.43 | 20.57±0.63 |
| SGADA [21] | 93.82±0.10 | **7.13**±0.22 | **91.70**±0.13 | **28.42**±0.34 | **16.10**±0.43 | 90.71±0.10 | **45.13**±0.32 | **17.26**±0.36 |
| SGPCS (ours) | **93.94**±0.04 | 7.16±0.16 | 91.55±0.13 | 28.52±0.21 | 16.16±0.26 | **91.57**±0.22 | 45.49±0.63 | 17.39±0.88 |
| UFLD-TO | 97.35 | 0.50 | 94.97 | 18.05 | 3.84 | 96.57 | 34.06 | 2.49 |
| ResNet-34 | LA | FP & FN | LA | FP | FN | LA | FP | FN |
| UFLD-SO | 90.35 | 22.25 | 89.42 | 32.35 | 21.19 | 89.17 | 48.86 | 23.67 |
| DANN [12] | 90.91±0.42 | 19.73±1.51 | 91.06±0.14 | 30.17±0.20 | 18.54±0.25 | 88.76±0.22 | 48.93±0.47 | 24.16±0.89 |
| ADDA [13] | 92.39±0.26 | 12.17±0.84 | 91.39±0.16 | 28.76±0.30 | 16.63±0.36 | 90.22±0.39 | 45.84±0.54 | 19.49±0.90 |
| SGADA [21] | 93.31±0.10 | 9.41±0.16 | 92.04±0.09 | 28.18±0.20 | 15.99±0.24 | **91.63**±0.03 | **44.18**±0.12 | **16.23**±0.16 |
| SGPCS (ours) | **93.53**±0.25 | **8.24**±0.91 | **93.29**±0.18 | **25.68**±0.48 | **12.73**±0.59 | 91.55±0.17 | 44.75±0.28 | 16.41±0.44 |
| UFLD-TO | 97.21 | 0.30 | 94.43 | 20.74 | 7.20 | 96.54 | 33.76 | 2.03 |

Figure 4: t-SNE visualization of MuLane dataset. The source domain is marked in blue, the real-world model vehicle target domain in red, and the TuSimple domain in green.

annotations in many cases but are not able to achieve comparable results to the supervised baseline (UFLD-TO).

In summary, we find quantitatively and qualitatively that the examined domain adaptation methods do not significantly improve the performance of lane detection and feature adaptation. For this reason, we believe that the proposed benchmark could facilitate the exploration of new domain adaptation methods to overcome these problems.

## 6 Conclusion

We present CARLANE, the first UDA benchmark for lane detection. CARLANE was recorded in three domains and consists of three datasets: the single-target datasets MoLane and TuLane and the multi-target dataset MuLane, which is a balanced combination of both. Based on the UFLD model, we conducted experiments with different UDA methods on CARLANE and found that the selected methods are able to adapt the model to target domains slightly and consistently. However, none of the methods achieve comparable results to the supervised baselines. The most significant performance differences are noticeable in the high false positive and false negative rates of the UDA methods compared to the target-only baselines, which is even more pronounced in the MuLane multi-target task. These false-positive and false-negative rates can negatively impact autonomous driving functions since they represent misidentified and missing lanes. Furthermore, as shown in the t-SNE plots of Figure 4, the examined well-known domain adaptation methods have no significant effect on feature alignment. The current difficulties of the examined UDA methods to adequately align the source and target domains confirm the need for the proposed CARLANE benchmark. We believe that CARLANE eases the development and comparison of UDA methods for lane detection. In addition, we open-source all tools for dataset creation and labeling and hope that CARLANE facilitates future research in these directions.

**Limitations.** One limitation of our work is that we only use a fixed set of track elements within our 1/8th scaled environment. These track elements represent only a limited number of distinct curve radii. Furthermore, neither buildings nor traffic signs exist in MoLane's model vehicle target domain.

|  | MoLane | TuLane | MuLane |
|---|---|---|---|

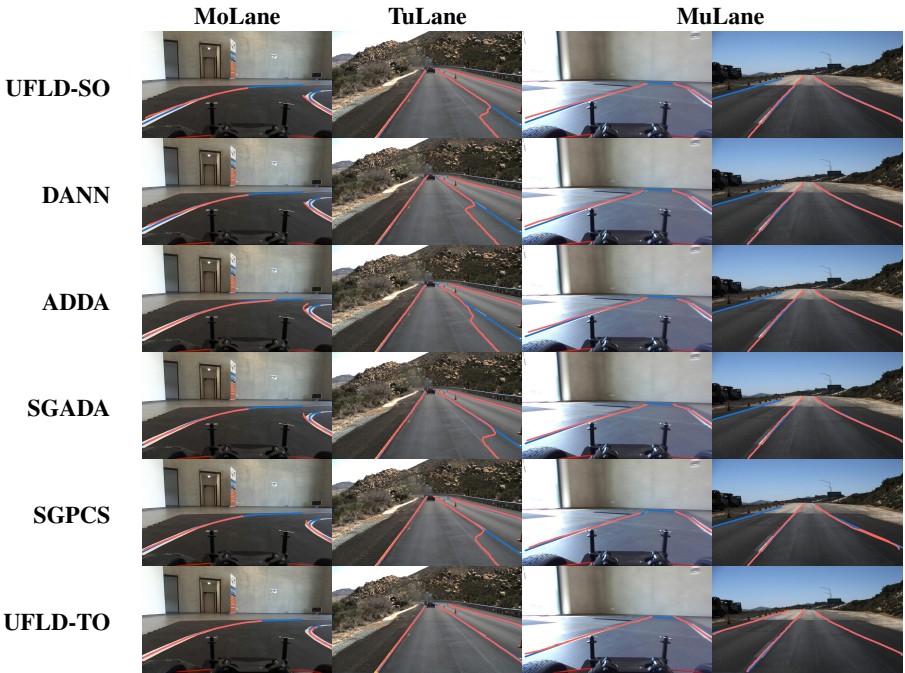

Figure 5: Qualitative results of target domain predictions. Ground truth lane annotations are marked in blue, predictions in red.

Moreover, the full-scale real-world target domain of TuLane is derived from TuSimple. TuSimple's data was predominantly collected under good and medium conditions and lacks variation in weather and time of day. In addition, we want to emphasize that collecting data for autonomous driving is still an ongoing effort and that datasets such as TuSimple do not cover all possible real-world driving scenarios to ensure safe, practical use. For the synthetically generated data, we limited ourselves to using existing CARLA maps without defining new simulation environments. Despite these limitations, CARLANE serves as a supportive dataset for further research in the field of UDA.

**Ethical and Responsible Use.** Considering the limitations of our work, UDA methods trained on TuLane and MuLane should be tested with care and under the right conditions on a full-scale car. However, real-world testing with MoLane in the model vehicle domain can be carried out in a safe and controlled environment. Additionally, TuLane contains open-source images with unblurred license plates and people. This data should be treated with respect and in accordance with privacy policies. In general, our work contributes to the research in the field of autonomous driving, in which a lot of unresolved ethical and legal questions are still being discussed. The step-by-step testing possibility across three domains makes it possible for our benchmark to include an additional safety mechanism for real-world testing.

## Acknowledgments and Disclosure of Funding

The authors would like to thank all anonymous reviewers for their helpful comments and recommendations, which contributed to the quality of the paper. This work was supported exclusively by the University of Applied Sciences Kempten.

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
