# OpenReview forum: "CARLANE: A Lane Detection Benchmark for Unsupervised Domain Adaptation from Simulation to multiple Real-World Domains"
_NeurIPS.cc/2022/Track/Datasets_and_Benchmarks — NeurIPS 2022 Datasets and Benchmarks _

### Official Review · Reviewer_9RHf · 2022-07-26
**Interesting paper on a lane detection benchmark**

**Rating:** 7
**Confidence:** 4
**Correctness:** Yes.
**Clarity:** Yes, it's well-written.

**Strengths:**

1. The paper proposes a systematic way to evaluate UDA methods on the proposed datset, which is useful for research in lane detection.
1. Data from simulation, the model vehicle, and real-world scenarios allows flexible usage to learn lane detection models.
1. The paper extends the standard single-target UDA to multi-target UDA, which is closer to real-world applications.

**Weaknesses:**

It would be good to consider different weather conditions in the 3 proposed datasets.

**Additional Feedback:**

NA

**Documentation:**

Yes, the authors open source their data and code.

**Ethics:**

No ethical concerns.

**Relation To Prior Work:**

Yes, it's well-discussed.

**Summary And Contributions:**

The paper introduces a 3-way benchmark called CARLANE focused on simulation to real-world unsupervised domain adaptation with single- and multi-target. The paper also provides dataset tools for labeling images. The authors conduct extensive experiments and draw interesting conclusions.

---

> ### Author Response · Authors · 2022-08-11
> **Response to helpful comments from Reviewer 9RHf**
>
> Dear reviewer,
>
> thank you for your encouraging review and for accepting our work. We are glad that you consider our benchmark to be flexible and systematic for research towards unsupervised domain adaptation for lane detection, which was one of our main goals. Regarding the different weather conditions, we apologize that this information is not described clearly enough in the paper. We, therefore, added more details on this topic to the main text. For the simulation data used in all three datasets, the weather conditions are varied systematically by adapting parameters for cloud density, rain intensity, puddles, wetness, wind strength, and fog density. More details can be found in the main paper and the implementation. As our current model vehicle is not water-proof and drives indoors, the variations are limited to environmental lightning conditions and track surfaces. For the real-world target domain, we utilized the recordings of TuSimple, which were captured under different weather conditions.
>
> We hope the details are more thoroughly addressed in the main paper now and want to thank you for your comment.

---

### Official Review · Reviewer_keEe · 2022-07-27
**A useful large-scale lane detection dataset for domain adaptation**

**Rating:** 6
**Confidence:** 4
**Correctness:** I think the claims made in the submis…

**Strengths:**

1. Lane detection is a cutting-edge field, which has gained much attention in recent years. The vast amount of collected labeled data has greatly facilitated progress in this field. Also, a large benchmark for cross-domain lane detection can greatly promote the development of the field.
2. Extensive experimental results and visualization are provided. The code is also provided, making it easy to reproduce the result.
3. The paper is well organized and easy to understand.


**Weaknesses:**

1. This dataset is somewhat simple for domain adaptation, i.e. the domain shift is small. First, the source-only model performs well on the target domain data, and all three settings can approach 90% lane accuracy. Second, the DA method is not significantly improved compared to the source-only results. It seems that DA has no obvious effect. This phenomenon is more obvious in the t-sne visualization. Third, the gap between the source-only results and the target-only results is small, which makes the significance of this study very limited.
2. More explanations are needed why DANN performs worse than source only in Tab3. Why does DANN cause the phenomenon of negative transfer, is it caused by the distribution of the dataset or is it caused by the characteristics of lane detection.
3. Only sim to real adaptation is included. Benchmark for 'real to real' (e.g. model vehicle - > TuSimple) is also important.
4. The statistics of the datasets proposed by the authors and those of previous works need to be compared.
5. Some related works for cross-domain detection and self-supervised land detection methods are missing.

**Additional Feedback:**

Please see the 'Weakness'.

**Clarity:**

The paper is well organized and easy to understand.


**Documentation:**

The authors provide urls for their dataset and code. I think the documentations provided by the authors satisfy the requirements of the NIPS DATASET track.

**Ethics:**

I think there is no ethical concern.

**Relation To Prior Work:**

This work proposes a 3-way UDA dataset and also proposes a UDA benchmark for the first time, which makes this work a unique contribution. Furthermore, this work has discussed the few previous UDA datasets in related work. However, the statistics of the datasets proposed by the authors and those of previous works need to be compared.

**Summary And Contributions:**

The paper proposes a 3-way sim to real benchmark for cross-domain lane detection, including single- and multi-target UDA. The author use the simulation software CARLA simulator and a real-built 1/8th Model vehicle to collect synthetic data and real lane data, respectively. Based on the three datasets, the authors introduce the first sim-to-real domain adaptation benchmark for lane detection, CARLANE. Well Known UDA methods like DANN and ADDA are evaluated, and the authors also propose a new method, named SGPCS, which achieves state-of-the-art performance on CARLANE.

---

> ### Author Response · Authors · 2022-08-11
> **Response to helpful comments from Reviewer keEe**
>
> Dear reviewer,
>
> we would like to thank you for your detailed and helpful review and for accepting our paper.
> We believe that the overall quality of the work has improved by incorporating your recommendations.
> In the following, we address your five points and how we have incorporated them into the paper.
>
> 1.  Although the source-only models perform relatively okay on the target tasks, when it comes to the accuracy values, we want to emphasize that the Lane Accuracy metric alone does not show the entire picture. The false-positive and false-negative values are significantly higher for the source-only models and domain adaption methods than for the trained target-only models. False-positive and false-negative values represent incorrectly detected and missing lanes, which can have a crucial impact on autonomous driving functions. As you already pointed out, the t-SNE visualizations only show a slight adaption toward the target domain. Therefore, we agree that the examined well-known domain adaptation methods have no significant effect on feature alignment and performance. For this reason, we believe that the proposed benchmark can help facilitate the research of new domain adaptation methods for lane detection to overcome these problems. We acknowledge that we addressed this issue too briefly in our paper and added a more detailed discussion of our benchmark results.
>
> 2. We have added an explanation of why DANN causes negative transfer in some of our domain adaptation tasks to our main text. Firstly, we want to point out that we are not the first to observe negative transfer of DANN for complex target tasks. Negative transfers also occur in tasks examined, for example, in [1-4]. A very detailed explanation of why DANN suffers from negative transfer can be found in [4]. The paper states, "The limitation [negative transfer] of DANN results from the unnecessary assumption that all source samples are equally useful." Although we balanced the source labels with the target domain per curve radius intervals, we do not have concrete labels classes. This results in source labels that do not perfectly match the target labels since some curve radii are observed only in the source domain but fall within one of the intervals specified for data sampling. This can be a problem for classical methods such as DANN. As stated in the aforementioned paper, "Relying on such 'unrelated' source samples can hurt the performance, leading to negative transfer". In Figure 3 of our work, we see that the labels of the MoLane source domain are widespread and contain samples that are not found in the target domain. In Table 3, we see a negative transfer of DANN. For the TuLane source domain, the labels in Figure 3 are not as widespread, and DANN does not suffer from negative transfer to MoLane in Table 3. It is important to point out that the ability of the underlying model and the learned feature space also play a role in the negative transfer, which may explain why our larger DANN model does not suffer from negative transfer to MoLane. One solution to mitigate this problem is the importance weighting of the source samples proposed by [4]. In summary, the negative transfer in our case can be explained by the distribution of the datasets.
>
> 3. Although we focused on sim-to-real transfer in this paper, we agree, that real-to-real is important too. One could use our dataset for real-to-real transfer from the model vehicle to the full-scale vehicle or vice versa. Therefore we added this scope to the Dataset Tasks section of the main paper and the datasheet in the appendix.
>
> 4. We have added Table 1 in the Appendix to compare our dataset with related work. In addition, we added Table 2 to compare the variations applied to generate synthetic data with related work, and Figure 3, which compares the visual quality of the collected simulation images with related work. We hope this improves the relation to prior work.
>
> 5. We have added about 10 cross-domain and self-supervised UDA references to the related work section.
>
> We would like to thank you again for your time and valuable review and hope that our revision addresses your concerns and increases confidence in your acceptance of our work.
>
> [1] FAN, Hehe, et al. Self-Supervised Global-Local Structure Modeling for Point Cloud Domain Adaptation With Reliable Voted Pseudo Labels. In: Proceedings of the IEEE/CVF Conference on Computer Vision and Pattern Recognition. 2022. S. 6377-6386.
> [2] TANWISUTH, Korawat, et al. A prototype-oriented framework for unsupervised domain adaptation. Advances in Neural Information Processing Systems, 2021, 34. Jg., S. 17194-17208.
> [3] KIM, Donghyun, et al. Cross-domain self-supervised learning for domain adaptation with few source labels. arXiv preprint arXiv:2003.08264, 2020.
> [4] WANG, Zirui, et al. Characterizing and avoiding negative transfer. In: Proceedings of the IEEE/CVF conference on computer vision and pattern recognition. 2019. S. 11293-11302.

---

### Official Review · Reviewer_gk58 · 2022-07-28
**Good work while some details need to be improved**

**Rating:** 7
**Confidence:** 2
**Correctness:** Yes.
**Clarity:** Well written.

**Strengths:**

Compensates for data scarcity and encourages future research in lane detection.

**Weaknesses:**

Experiments part are limited. The algorithms covered by this paper is not well rounded. More classic algorithm should be included.

**Additional Feedback:**

No.

**Documentation:**

Yes.

**Ethics:**

No.

**Relation To Prior Work:**

Clearly discussed.

**Summary And Contributions:**

This paper proposes CARLANE, a 3-way sim-to-real domain adaptation benchmark for 2D lane detection, which encompasses the single-target datasets MoLane, TuLane and the multi-target dataset MuLane.

---

> ### Author Response · Authors · 2022-08-11
> **Response to helpful comments from Reviewer gk58**
>
> Dear reviewer,
>
> first of all, we would like to thank you for your review and for accepting our paper. We are glad that you share our point of view that our dataset compensates for data scarcity and promotes future research in lane detection. For the classical domain adaptation algorithm, we train DANN, which was first proposed in 2015. Since seven years is a long time in deep learning, we hope this algorithm can be considered a classical approach.

---

### Author Response · Authors · 2022-08-11
**Paper and Supplementary Material Revision**

We thank all reviewers for their helpful comments and recommendations and would like to inform you that we have uploaded a revision of our main paper and supplementary materials, as well as responses to your reviews. We hope that the revision adequately addresses your concerns and builds further confidence in your acceptance of our work.

---

### Meta-Review · Area_Chair_Dhjn · 2022-09-02

**Recommendation:** Accept
**Confidence:** 4

**Metareview:**

The meta reviewer has read the paper, rebuttal, reviews, and the author's revision. The meta reviewer agrees with the reviewers that this lane benchmark is of interest to UDA, and the authors release the data, tool, and source codes to support their claimed contribution. The meta reviewer thus recommends acceptance.

---

### Decision · Program_Chairs · 2022-09-16

Accept